# Bone mineral density reference standards for Chinese children aged 3–18: cross-sectional results of the 2013–2015 China Child and Adolescent Cardiovascular Health (CCACH) Study

Junting Liu,[1] Liang Wang,[2] Jinghui Sun,[3] Gongshu Liu,[4] Weili Yan,[5] Bo Xi,[6] Feng Xiong,[7] Wenqing Ding,[8] Guimin Huang,[1] Steven Heymsfield,[9] Jie Mi[1]

► Prepublication history and additional material is available. To view please visit the journal (http://dx.doi.org/10.1136/bmjopen-2016-014542).

JL and LW contributed equally. JS, GL, WY, BX, FX and WD contributed equally.

## ABSTRACT

**Objectives** No nationwide paediatric reference standards for bone mineral density (BMD) are available in China. We aimed to provide sex-specific BMD reference values for Chinese children and adolescents (3–18 years).

**Methods** Data (10 818 participants aged 3–18 years) were obtained from cross-sectional surveys of the China Child and Adolescent Cardiovascular Health in 2015, which included four municipality cities and three provinces. BMD was measured using Hologic Discovery Dual Energy X-ray Absorptiometry (DXA) scanner. The DXA measures were modelled against age, with height as an independent variable. The LMS statistical method using a curve fitting procedure was used to construct reference smooth cross-sectional centile curves for dependent versus independent variables.

**Results** Children residing in Northeast China had the highest total body less head (TBLH) BMD while children residing in Shandong Province had the lowest values. Among children, TBLH BMD was higher for boys as compared with girls; but, it increased with age and height in both sexes. Furthermore, TBLH BMD was higher among US children as compared with Chinese children. There was a large difference in BMD for height among children from these two countries. US children had a much higher BMD at each percentile (P) than Chinese children; the largest observed difference was at P50 and P3 and the smallest difference was at P97.

**Conclusions** This is the first study to present a sex-specific reference dataset for Chinese children aged 3–18 years. The data can help clinicians improve interpretation, assessment and monitoring of densitometry results.

## INTRODUCTION

Osteoporosis, as a widespread metabolic bone disease, has become a global public health problem. More than 200 million people worldwide and 10 million people in the US were afflicted with osteoporosis in 2013.[1] In China, the population with osteoporosis in 1997 was 83.9 million, which is projected to increase to 212 million by

### Strengths and limitations of this study

► It is the first study to present a sex-specific reference dataset for Chinese children aged 3–18 years.
► A number of reference curves were developed using objective measures from DXA.
► The large sample size enabled us to construct stable reference curves and extended previous studies though covering a wide range of ages: from 3 to 18 years.
► The data collected were not nationally representative; hence, we could not generalise the results to the entire Chinese paediatric population.
► Our developed reference curve could not be used alone as a clinical criteria for the diagnosis of paediatric osteoporosis.

2050.[2] The burdens associated with osteoporosis are heavy, with the most common being fractures. It has been estimated that the treatment cost for hip fracture, in China, would be more than 85 billion Chinese Yuan (ie, about US$13 billion) in 2020, and 1800 billion Chinese Yuan (ie, about US$271 billion) in 2050.[3]

Peak bone mass is an important determinant for osteoporotic fracture risk. A higher peak bone mass is associated with lower risk of osteoporotic fractures later in life.[4] In general, individuals have peak bone mass in their early 20s; however, it may occur earlier in women.[5 6] Thus, investigating the pattern of bone mineralization in children is of great interest. Dual-energy X-ray absorptiometry (DXA) has been recommended by the International Society for Clinical Densitometry (ISCD)[7] as a current criterion for measuring body mineral content (BMC) and bone mineral density (BMD). This measurement method has high precision, accuracy,

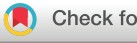

For numbered affiliations see end of article.

**Correspondence to**
Dr Jie Mi; Jiemi@vip.163.com

speed, low radiation exposure, ease of use and a strong correlation with established methods.[8–14]

Paediatric osteoporosis is defined as having a history of pathological fractures and low bone mineral content or density.[15] To diagnose paediatric osteoporosis, DXA is the most widely used bone densitometry technique in children.[7] Although 60% of peak bone mass is genetically determined, the increase of bone mass can be influenced by other factors, including intake of dietary products (eg, calcium and vitamin D) and weight-bearing physical activity (eg, jumping).[16]

While diagnosing children can be problematic due to a lack of a widely used reference, the Bone Mineral Density in Childhood Study has developed an online BMD z-score calculator for healthcare providers.[17]. Widespread reference for multiple diagnostic measurements is important to ensure correct diagnosis among children so that treatment and management for osteoporosis can begin. Uniformity in diagnosis is also imperative for public health so that epidemiological studies can be done across geographic regions, ethnicities and social economic status to properly inform prevention practices.

For children, DXA can be used to measure whole body bone area ($cm^2$) and whole body BMC (g), leading to BMD (ie, BMC divided by bone area ($g/cm^2$)), which indicates the paediatric bone status. The use and correct identification of indicators for BMD is particularly important; this requires appropriate reference data that are matched for multiple factors including sex,[18–20] chronological age, height, weight, pubertal development and ethnicity.[18 21–23] The preferred skeletal sites are lumbar spine (L1–L4) and total body, excluding the head.[24] The cranium should not be included for the total body scan analysis, because the head constitutes a large portion of the total body bone mass but changes little with growth, activity or disease. Thus, including the skull is likely to mask gains or losses at other skeletal sites.[25]

Childhood and adolescence are critical periods for bone growth, with half of peak bone mass developed in childhood. If BMD could be evaluated in childhood and adolescence, and children with low BMD could be intervened on, then peak bone mass can be improved in adulthood. Subsequently, osteoporosis occurrence can be delayed, and severity of osteoporosis can be reduced. A number of studies have shown that optimal BMC and BMD in childhood and adolescence would reduce risk of osteoporotic fracture several decades later.[26] To date, no paediatric reference standards for BMD are available in China. Furthermore, utilisation of US measures based on reference values derived from the US National Health and Nutrition Examination Survey (NHANES) data might not be applicable for Chinese paediatric populations. Therefore, it is critically important and urgent to develop reference values of BMD for Chinese children based on data gathered from Chinese children. These reference standards can help clinicians have a better estimation of the percentile on which Chinese children's DXA-measured bone

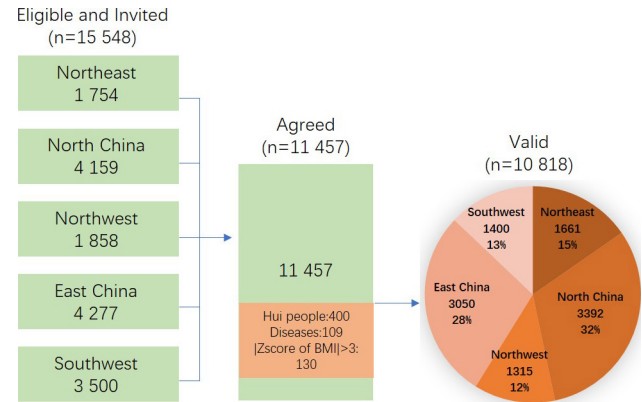

**Figure 1** Flow chart of the participants' recruitment, China Child and Adolescent Cardiovascular Health (CCACH) Study.

parameter falls in reference to their age or height. The aim of this study was twofold: (1) to develop sex-specific reference percentile curves for Chinese children based on a large, nationwide sample; and (2) to compare our findings with those of the study conducted by Kelly and colleagues[27] using US NHANES data.

## METHODS
### Participants
The American Heart Association defined a new concept of 'ideal cardiovascular health (CVH)' for American adults and children in 2010,[28] which included four health behaviours (non-smoking, physically active, normal body mass index (BMI) and healthy diet) and three health factors (normal blood pressure, total cholesterol and fasting glucose). Based on these seven health behaviours and factors, the construct of CVH comprehensively combines the traditional risk factors proposed by the Framingham Heart Study, as well as lifestyle behaviours, emphasising the importance of overall strategies in achieving optimal CVH. Our data were obtained from the China Child and Adolescent Cardiovascular Health (CCACH) Study. It is a large, nationwide, ongoing study designed to examine CVH, BMD, body composition and daily sodium intake among Chinese children aged 3–18 (figure 1). A stratified cluster random sampling method was used to extract 100 children per age group in each cluster (50 boys and 50 girls). Because the majority of children were Han, accounting for 90% of the total Chinese paediatric population, CCACH study mainly collected CVH data among Han children. Children were recruited from seven provinces (centres) covering geographical areas in North, East, Northeast, Northwest and Southwest regions of China, across latitude 29.6° north to latitude 41.7° north (see online supplementary figure 1). Eight exclusion criteria were: (1) the inability to give informed consent; (2) any condition, or use of any drug known to affect bone health; (3) non-removable objects (eg, prostheses, implants or casts); (4) body weight greater than 204 kg, height greater than 197.5 cm or weight or height beyond the measurement range of

machines; (5) participation in research involving ionising radiation in the past year; (6) no history of bone fracture; (7) some chronic diseases and (8) pregnancy. Among the 15 548 eligible participants, 11 057 agreed to participate and signed written informed consent documents. Data were collected on 11 457 children; of these, 400 Hui children were excluded, 109 were excluded due to kidney or thyroid disease, and 130 were excluded because of implausible values of weight (light or heavy, BMI ≥3 SDs). Consequently, 10 818 children (5509 males and 5309 females) were used in the final analysis.

The study was approved by the Research Ethics Committee of Capital Institute of Pediatrics of China. The written informed consent documents for children were obtained from their parents or guardians.

### DXA acquisition and quality control

The whole-body scans were performed using Hologic Discovery (A, W and Wi) fan-beam densitometers (Hologic, Bedford, Massachusetts, USA). The coefficient of variation (CV%) was used as the quality control procedure. CV% of A, W and Wi was 0.471%, 0.302% and 0.358%, respectively. Following the recommendations of the ISCD, a high level of quality control was maintained throughout the DXA data collection and scan analysis. The specific procedures were as follows: (1) Standard phantom was checked before scanning the participants and calibrating of the DXA machines every morning; (2) an operator at each study location was trained and the same technician certified by ISCD administered DXA procedures. The training materials included the ISCD's official technician hands-on training materials and the manufacturer's handbook including testing procedures and operation methods; (3) at each centre, lumbar spine and hip joints of 15 participants were scanned three times for the calculation. After each scan, they had to leave the scanner to repose before the next scan. All our operator staff passed the precision quality control required by ISCD, which controlled the CV% to less than 1.9% for lumbar spine scans, 1.8% for hip joint scans and 2.5% for femoral neck scans (see table 1) and (4) all participants

were requested to remove outer garments and objects that would potentially interfere with testing. All DXA values were analysed using Hologic Apex software Version 4.0 following the manufacturer's guidelines.

### STATISTICAL ANALYSIS

The DXA measures for our paediatric participants were modelled against age or height as the independent variable. The LMS statistical method (LMS ChartMaker Pro Version 2.54)[29][30] using a curve fitting procedure was employed to construct reference smooth cross-sectional centile curves for BMD/body composition versus age or height. The LMS technique estimates three parameters: median (M), coefficient of variation (S) and power in the Box-Cox transformation (L). This summarises the distribution of the dependent variable given the covariate by L depending on the skewness of the distribution while M and S gives smooth curves for the dependence of L, M and S on the covariate. These three parameters vary as a function of independent variables fully describing the reference data. Once these parameters are estimated, then centile curves can be constructed using the formula that has been described elsewhere.[10][21] To avoid overfitting of the curves, parsimonious models were used instead of complex models when they had similar goodness of fit. After fitting the curves, the distribution of z-scores was observed and outlier analysis was performed to verify that all values were within physiological range (eg, observations ≥3 SDs were removed). All the analyses were performed using SAS Version 9.4. The z-scores were calculated by the following equation:

$$Z = \frac{\left(\frac{X}{M}\right)^{L} - 1}{LS}$$

If L=0,

$Z = \ln(X/M)/S$

Where X is the TBLH BMD, L is the power transformation, M is the median value and S is the population SD[24] Percentiles were obtained from z-scores, for example, z-scores of −1.645, −1.282, 0, 1.282 and 1.645 correspond to the 5th, 10th, 50th, 90th and 95th percentiles, respectively.

### RESULTS

Among the 10 818 children residing in seven geographically distributed centres across China, children from Beijing accounted for the largest proportion (16.2%), while children from Ningxia accounted for the smallest proportion (12.2%) (table 2). In addition, the age- and sex-specific TBLH BMD was significantly different among seven centres. Children residing in Northeast China had the highest TBLH BMD while children residing in Shandong Province had the lowest value (table 2).

TBLH BMD of children by sex and height is shown in table 3. TBLH BMD increased with age for both sexes, for example, from 0.412 g/cm² at age 3 to 0.948 g/cm² at

**Table 1** The distribution of coefficient of variation (CV, %) for lumbar spine, hip joint and femoral neck by study location, the China Child and Adolescent Cardiovascular Health (CCACH) Study (n=10 818)

| Province/Centre | Lumbar spine | Hip joint | Femoral neck |
|---|---|---|---|
| Beijing | 0.88 | 1.42 | 1.91 |
| Shanghai | 0.97 | 1.34 | 2.33 |
| Tianjin | 1.06 | 1.02 | 1.99 |
| Chongqing | 0.59 | 1.62 | 2.05 |
| Jilin | 1.58 | 1.79 | 1.39 |
| Shandong | 1.11 | 1.79 | 1.97 |
| Ningxia | 1.25 | 1.07 | 1.89 |

**Table 2** Geographical distribution of sample, The China Child and Adolescent Cardiovascular Health (CCACH) Study (n=10 818)

| Geographical area | Province/Centre | Latitude | n | % | TBLH BMD |
|---|---|---|---|---|---|
| Northeast | Jilin | N 40°52'~46°18' | 1661 | 15.4 | 0.785±0.160 |
| North China | Beijing | N39°26'~41°03' | 1739 | 16.1 | 0.734±0.151 |
| | Tianjin | N38°33'~40°15' | 1653 | 15.3 | |
| Northwest | Ningxia | N35°14'~39°23' | 1315 | 12.2 | 0.743±0.129 |
| East China | Shanghai | N30°40'~31°53' | 1613 | 14.9 | 0.740±0.131 |
| | Shandong | N34°25'~38°23' | 1437 | 13.3 | |
| Southwest | Chongqing | N28°10'~32°13' | 1400 | 12.9 | 0.778±0.143 |
| F* | | | | | 163.11 |
| p | | | | | <0.0001 |

*Analysis of covariance, adjusted for age and sex.
BMD, bone mineral density; TBLH, total body less head.

age 18 for boys, and from 0.399 g/cm$^2$ to 0.853 g/cm$^2$ for girls. Similarly, TBLH BMD also increased with height for boys and girls. Overall, TBLH BMD was higher in boys than in girls.

The percentiles of TBLH BMD among Chinese and US children were compared and are shown in figure 2. TBLH BMD increased with age among boys and girls in both countries; however, TBLH BMD was comparatively higher in children living in the US than those in China. Among boys, the differences at each percentile (ie, P3, P50 and P97) between China and the US were similar, in general; the differences were small in young ages and peaked at age 18. BMD at P3, P50 and P97 was higher in American boys than in Chinese boys. There was a gradual increase in BMD among girls and boys after age 14 and age 16, respectively. Furthermore, BMD for age was higher in boys than in girls in both countries.

In both China and the USA, BMD increased with height in both sexes; US children had a much higher BMD than Chinese children at each percentile, with the largest difference at P50 and P3 and the least difference at P97. The increasing rate of BMD for height was stable, with an obvious increase from height 155 to 175 cm among US children. Among girls with low heights (eg, <135 cm), BMD value at P3 in US children was similar to BMD value at P50 in Chinese children. A clear increase in BMD for height was observed from 150 to 165 cm among US girls as compared with increase in BMD from 145 to 155 cm among Chinese girls. At height greater than 155 cm, the difference in BMD became even larger. BMD peaked at 180 cm where BMD at P3 and P50 in US children were close to those at P50 and P97 in Chinese children. BMD for height was also higher in boys than in girls.

In each sex, LMS parameters and reference centiles of BMD by age, BMD by height, fat mass percentage (FMP) by age, fat mass index (FMI) by age and free fat mass index (FFMI) by age can be found in online supplementary tables 1–10.

## DISCUSSION

In this study, we developed sex-specific reference standards using Hologic Discovery DXA scanner in a large nationwide sample of Chinese children aged 3–18 years. In addition, we compared the results from this study with references used in NHANES data, and found that BMD was lower in Chinese children than in US children. To the best of our knowledge, this study is the first attempt to develop reference BMD values for Chinese children using a large nationwide paediatric population in China, and compare it with alternative available references.

China has a vast territory with large populations. Disparities in lifestyle behaviours, socioeconomic status, obesity and metabolic disorders may contribute to differences in BMD among children and adolescents. There is an urgent need to establish unified, nationwide BMD reference standards for children and adolescents, to help better evaluate their growth status. In addition, adequate peak bone mass at an early age was positively associated with lifelong health (eg, osteoporosis at an older age)[16 31]; therefore, an accurate assessment of BMD is essential in assisting development of future preventive methods to reduce the physical and economic burden of osteoporosis.

The difference in BMD among children may be due to their different dietary patterns. Compared with children consuming traditional Chinese diet (predominately consisting of vegetables and grains, and featuring plant-based protein, which is high in fibre, and low in cholesterol and fat), children in Western countries are more likely to consume foods that consists of more meats and milk. Thus their diet is high in energy density, fat and protein.[32] Different genetic backgrounds should also be an explanation for the difference in BMD among children in China and the US. There are no reference standards of BMD available for the Chinese paediatric populations based on a large dataset. The present study utilised all body and regional DXA data to establish age-, sex- and height-specific reference values for Chinese children

**Table 3** TBLH BMD (g/cm$^2$) of children aged 3 to 18 years by sex and height, The China Child and Adolescent Cardiovascular Health (CCACH) Study (n=10 818)

| Age, height | Male | | | | | | | Female | | | | | | |
|---|---|---|---|---|---|---|---|---|---|---|---|---|---|---|
| | Sample size | Mean±SD | TBLH BMD percentiles | | | | | Sample size | Mean±SD | TBLH BMD percentiles | | | | |
| | | | 5th | 10th | 50th | 90th | 95th | | | 5th | 10th | 50th | 90th | 95th |
| **Age (years)** | | | | | | | | | | | | | | |
| 3 | 110 | 0.412±0.030 | 0.350 | 0.360 | 0.395 | 0.434 | 0.446 | 122 | 0.399±0.025 | 0.349 | 0.357 | 0.385 | 0.417 | 0.426 |
| 4 | 269 | 0.444±0.037 | 0.373 | 0.383 | 0.421 | 0.463 | 0.476 | 267 | 0.426±0.032 | 0.366 | 0.374 | 0.406 | 0.441 | 0.452 |
| 5 | 322 | 0.479±0.039 | 0.409 | 0.420 | 0.463 | 0.511 | 0.526 | 259 | 0.470±0.036 | 0.401 | 0.411 | 0.449 | 0.492 | 0.505 |
| 6 | 408 | 0.531±0.046 | 0.445 | 0.458 | 0.505 | 0.559 | 0.576 | 376 | 0.515±0.047 | 0.434 | 0.446 | 0.491 | 0.542 | 0.558 |
| 7 | 401 | 0.582±0.050 | 0.489 | 0.503 | 0.557 | 0.618 | 0.636 | 359 | 0.557±0.049 | 0.468 | 0.482 | 0.535 | 0.594 | 0.612 |
| 8 | 399 | 0.610±0.051 | 0.523 | 0.538 | 0.596 | 0.663 | 0.684 | 333 | 0.595±0.055 | 0.498 | 0.514 | 0.575 | 0.642 | 0.662 |
| 9 | 332 | 0.644±0.052 | 0.547 | 0.564 | 0.627 | 0.699 | 0.721 | 340 | 0.629±0.061 | 0.526 | 0.544 | 0.612 | 0.687 | 0.710 |
| 10 | 345 | 0.673±0.059 | 0.571 | 0.589 | 0.656 | 0.733 | 0.757 | 303 | 0.670±0.072 | 0.556 | 0.576 | 0.652 | 0.734 | 0.758 |
| 11 | 327 | 0.709±0.068 | 0.597 | 0.615 | 0.687 | 0.770 | 0.795 | 293 | 0.722±0.071 | 0.593 | 0.615 | 0.696 | 0.784 | 0.810 |
| 12 | 352 | 0.748±0.071 | 0.628 | 0.648 | 0.725 | 0.814 | 0.842 | 290 | 0.765±0.069 | 0.635 | 0.658 | 0.743 | 0.834 | 0.861 |
| 13 | 319 | 0.798±0.075 | 0.665 | 0.687 | 0.771 | 0.867 | 0.898 | 274 | 0.800±0.071 | 0.673 | 0.696 | 0.781 | 0.873 | 0.900 |
| 14 | 299 | 0.845±0.075 | 0.704 | 0.728 | 0.818 | 0.923 | 0.956 | 298 | 0.818±0.061 | 0.701 | 0.724 | 0.807 | 0.897 | 0.924 |
| 15 | 440 | 0.877±0.085 | 0.737 | 0.762 | 0.858 | 0.971 | 1.006 | 475 | 0.829±0.062 | 0.720 | 0.741 | 0.823 | 0.911 | 0.937 |
| 16 | 540 | 0.916±0.096 | 0.763 | 0.789 | 0.891 | 1.011 | 1.048 | 571 | 0.843±0.068 | 0.734 | 0.755 | 0.835 | 0.922 | 0.949 |
| 17 | 405 | 0.953±0.094 | 0.791 | 0.819 | 0.927 | 1.054 | 1.094 | 456 | 0.853±0.068 | 0.745 | 0.766 | 0.845 | 0.933 | 0.960 |
| 18 | 241 | 0.948±0.093 | 0.808 | 0.836 | 0.949 | 1.081 | 1.123 | 293 | 0.853±0.068 | 0.751 | 0.771 | 0.850 | 0.938 | 0.966 |
| **Height (cm)** | | | | | | | | | | | | | | |
| 100 | 126 | 0.401±0.025 | 0.366 | 0.372 | 0.395 | 0.422 | 0.431 | 171 | 0.396±0.024 | 0.360 | 0.366 | 0.390 | 0.417 | 0.425 |
| 105 | 169 | 0.434±0.024 | 0.386 | 0.393 | 0.420 | 0.452 | 0.462 | 200 | 0.428±0.025 | 0.382 | 0.389 | 0.416 | 0.447 | 0.457 |
| 110 | 227 | 0.463±0.028 | 0.410 | 0.418 | 0.449 | 0.486 | 0.498 | 175 | 0.459±0.030 | 0.405 | 0.413 | 0.443 | 0.480 | 0.492 |
| 115 | 275 | 0.504±0.037 | 0.437 | 0.447 | 0.483 | 0.526 | 0.540 | 278 | 0.492±0.035 | 0.431 | 0.439 | 0.474 | 0.519 | 0.534 |
| 120 | 313 | 0.538±0.044 | 0.467 | 0.478 | 0.519 | 0.570 | 0.586 | 284 | 0.534±0.042 | 0.458 | 0.468 | 0.509 | 0.563 | 0.582 |
| 125 | 358 | 0.577±0.045 | 0.497 | 0.509 | 0.555 | 0.613 | 0.632 | 336 | 0.566±0.050 | 0.487 | 0.499 | 0.546 | 0.611 | 0.634 |
| 130 | 363 | 0.604±0.046 | 0.524 | 0.537 | 0.588 | 0.652 | 0.674 | 299 | 0.600±0.053 | 0.513 | 0.526 | 0.579 | 0.653 | 0.679 |
| 135 | 330 | 0.642±0.053 | 0.550 | 0.564 | 0.620 | 0.691 | 0.714 | 256 | 0.630±0.057 | 0.534 | 0.548 | 0.607 | 0.689 | 0.718 |
| 140 | 312 | 0.667±0.054 | 0.573 | 0.588 | 0.649 | 0.726 | 0.752 | 254 | 0.662±0.068 | 0.560 | 0.576 | 0.642 | 0.732 | 0.764 |
| 145 | 258 | 0.692±0.064 | 0.595 | 0.611 | 0.677 | 0.761 | 0.789 | 277 | 0.722±0.079 | 0.597 | 0.615 | 0.690 | 0.787 | 0.821 |
| 150 | 263 | 0.726±0.068 | 0.620 | 0.637 | 0.708 | 0.798 | 0.829 | 489 | 0.784±0.078 | 0.645 | 0.665 | 0.747 | 0.848 | 0.882 |
| 155 | 254 | 0.769±0.075 | 0.651 | 0.669 | 0.745 | 0.845 | 0.878 | 830 | 0.813±0.070 | 0.688 | 0.709 | 0.793 | 0.892 | 0.923 |
| 160 | 336 | 0.828±0.087 | 0.686 | 0.706 | 0.788 | 0.897 | 0.934 | 838 | 0.829±0.065 | 0.718 | 0.739 | 0.821 | 0.913 | 0.942 |
| 165 | 514 | 0.859±0.090 | 0.721 | 0.742 | 0.831 | 0.948 | 0.989 | 444 | 0.855±0.066 | 0.741 | 0.762 | 0.843 | 0.932 | 0.959 |

Continued

**Table 3** Continued

| Age, height | Male | | | | | | | | Female | | | | | | | |
| --- | --- | --- | --- | --- | --- | --- | --- | --- | --- | --- | --- | --- | --- | --- | --- | --- |
| | Sample size | Mean±SD | TBLH BMD percentiles | | | | | | Sample size | Mean±SD | TBLH BMD percentiles | | | | | |
| | | | 5th | 10th | 50th | 90th | 95th | | | | 5th | 10th | 50th | 90th | 95th | |
| 170 | 670 | 0.896±0.086 | 0.753 | 0.776 | 0.871 | 0.997 | 1.040 | | 120 | 0.878±0.076 | 0.761 | 0.783 | 0.864 | 0.953 | 0.980 | |
| 175 | 432 | 0.930±0.095 | 0.784 | 0.808 | 0.908 | 1.041 | 1.088 | | – | – | – | – | – | – | – | |
| 180 | 222 | 0.977±0.103 | 0.813 | 0.838 | 0.943 | 1.084 | 1.134 | | – | – | – | – | – | – | – | |

BMD, bone mineral density; TBLH, total body less head.

aged 3–18 years, and provided centile curves of BMD, which can be used in both research and clinical settings.

DXA, a widely available technology, is capable of providing regional measures of fat and lean mass. The distribution of fat and lean mass is likely to predict health outcomes,[27] as shown previously in an adult sample.[30] DXA measurements provide precise body composition analysis with a low radiation dose,[33] and are able to detect small changes in body composition. DXA devices from different manufacturers might not give identical results due to differences in calibration and bone edge detection algorithms.[34] Therefore, to calculate reliable results, one should use reference data obtained with an identical DXA device. For instance, US NHANES data employed Hologic (Hologic, Bedford, Massachusetts) and GE-Lunar as the two dominant DXA manufacturers. Despite the similar DXA technology used for by manufacturers, the results of BMD and body composition can be different due to proprietary techniques used, algorithms to calculate the BMD, as well as the regions of interest. This leads to various DXA values for patients who are scanned by different DXA systems.[8] To avoid the comparability problem, our study used Hologic as the only manufacturer. We used our collected data on BMD percentiles among Chinese children to compare with those among US children (BMD values were obtained from US NHANES data).[27]

Compared with our study, another study,[35] conducted among southern Chinese paediatric populations, was different in selected geographic characteristics, sample size and age range. For example, although 900 children (5–19 years) came from Guangzhou (Southern China) and about 600 (14–19 years) came from Zhejiang (East China), the study did not consider variation in geographic characteristics; and their age range was smaller than ours (3–18 years). Equipment used in that study was GE Lunar, therefore their results cannot be compared with ours, having used Hologic DXA.

Our study has many strengths. First, we developed a number of reference curves generated from objective measures from DXA including sex-age-specific and sex-height-specific TBLH BMD, providing nationwide references for Chinese children. Second, the large sample size enabled us to construct stable reference curves and extended previous studies though covering a wide range of ages from 3 to 18 years. Third, our study followed the recommendation of ISCD and studied TBLH BMD for Chinese children.

Several limitations are noteworthy. First, though our data is nationally and geographically distributed across China, it is not nationally representative; hence, the results may not be generalisable to the entire Chinese paediatric population. For example, although Ningxia Province, representative of Northwest China, has thousands of Hui population, we mainly collected data from schools including Han population. Thus, our sampled participants across China likely only represent Chinese Han children. Moreover, our developed reference curve could not be used alone as a clinical criteria for the diagnosis of osteoporosis. This is because:

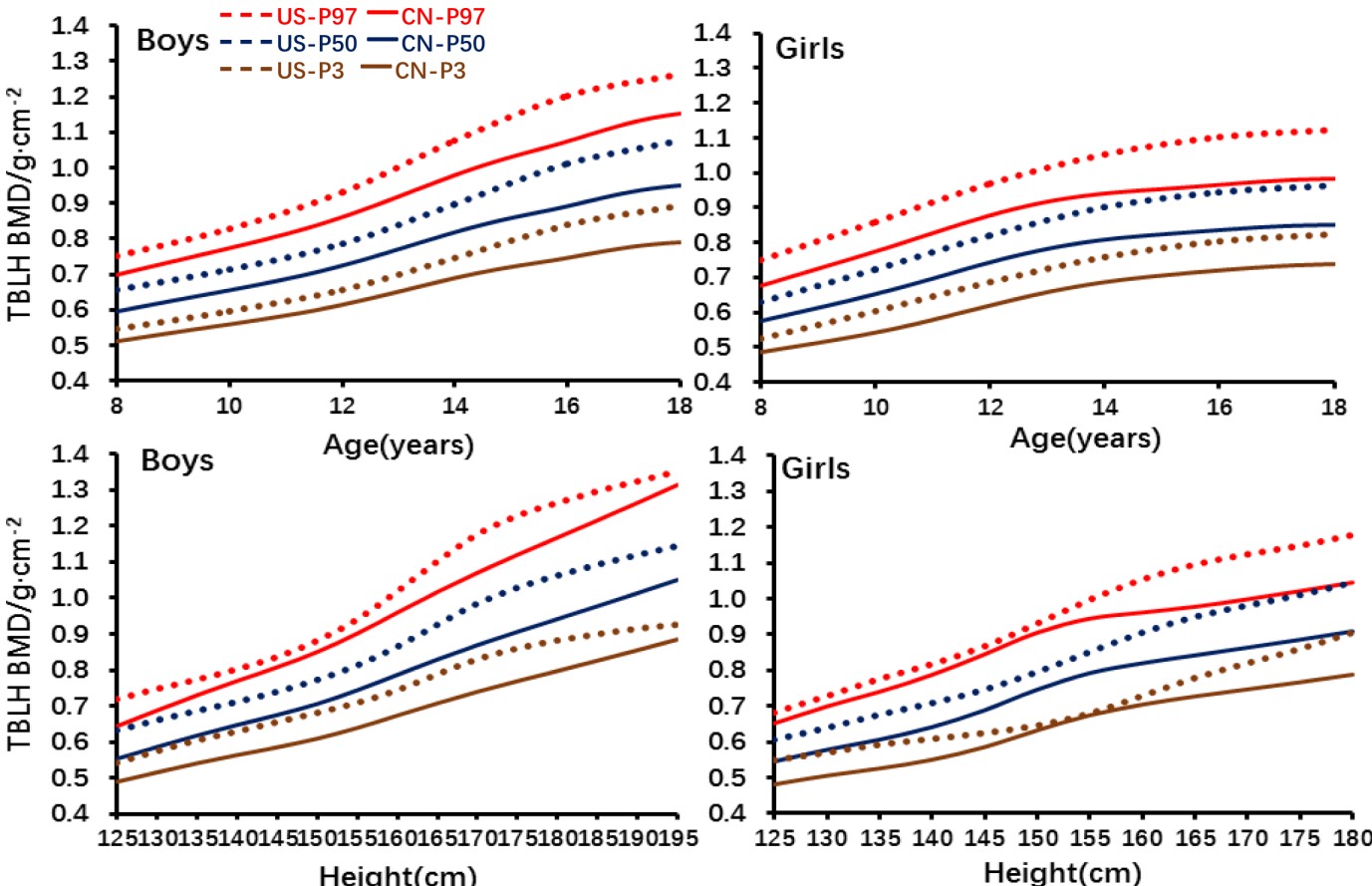

**Figure 2** The percentiles of TBLH BMD among Chinese and US children by age and height. CN, data from China; TBLH BMD, total body less head bone mineral density; US, data from NHANES.

(1) low bone mass may reflect only a small part of the true burden of osteoporosis and (2) BMD could not demonstrate other important components of bone strength. Despite these limitations, there are some important public health implications. First, the BMD percentile can be used to precisely evaluate child and adolescent growth, and help high-risk populations become connected with better interventions. Second, since childhood obesity has been epidemic in the nation, assessing BMD can facilitate association studies between body composition (eg, fat and muscle) and bone health, to better determine health status for obese individuals. Furthermore, it can be used for evaluating effectiveness of sports intervention among children. Clinical implications are also notable. First, childhood kidney disease, rheumatism and immune diseases, and bone and joint diseases can directly and indirectly influence bone growth; Second, BMD tests can evaluate these diseases' adverse outcomes on an individual's health status and treatment effects. In addition, antiepileptic drugs and other hormonal drugs can affect osteogenesis, and BMD reference can be used as a standard for evaluation of treatment effects and side effects for children.

## CONCLUSION

This is the first study to develop sex-specific references among Chinese children aged 3–18 years using the Hologic Discovery DXA scanner. The information can be used to guide clinicians and improve interpretation, assessment and monitoring of BMD results for Chinese children. However, it needs to be combined with other indicators (eg, blood pressure, serum lipid and calcium-phosphorus metabolism) to identify the diagnosis cut points for osteopenia.

**Author affiliations**
[1]Department of Epidemiology, Capital Institute of Pediatrics, Beijing, China
[2]Department of Biostatistics and Epidemiology, College of Public Health, East Tennessee State University, Johnson City, Tennessee, USA
[3]Department of Cardiovascular Medicine, The First Hospital of Jilin University, Changchun, China
[4]Project Office, Tianjin Women's and Children's Health Center, Tianjin, China
[5]Department of Clinical Epidemiology, Children's Hospital of Fudan University, Shanghai, China
[6]Department of Public Health, Shandong University, Jinan, China
[7]Department of Endocrinology, Children's Hospital of Chongqing Medical University, Chongqing, China
[8]Department of Public Health, Ningxia Medical University, Yinchuan, China
[9]Department of Metabolism-Body Composition, Pennington Biomedical Research Center, Baton Rouge, Louisiana, USA

**Correction notice** This article has been corrected since it first published. The equal contributors statement has been corrected.

**Contributors** JL, LW contributed equally, and they are first co-authors. JS, GL, WY, BX, FX and WD contributed equally, and they are second co-authors. JM had full access to all the data in the study and takes responsibility for the integrity of the data and the accuracy of the data analysis. Study concept and design: JM. Acquisition, analysis or interpretation of data: JS, GL, WY, BX, FX, WD, GH, JM.

Drafting of the manuscript: JL, LW. Critical revision of the manuscript for important intellectual content: JL, LW, SH, JM. Statistical analysis: JL, LW. Obtained funding: JM. Administrative, technical or material support: JL, LW, WY, BX, FX, WD, JM. Study supervision: JM.

**Funding** The study was supported by grants from the 'Twelfth Five Year Plan' of the China National Science and Technology (2012BAI03B03), the Beijing Health System Leading Talent Grant (2009108), the Beijing Training Project for the Leading Talents in Science and Technology (2011LJ07).

**Disclaimer** The content is solely the responsibility of the authors and does not necessarily represent the official views of the funders. The funding sources had no role in the design and conduct of the study; collection, management, analysis and interpretation of the data; preparation, review or approval of the manuscript; and decision to submit the manuscript for publication.

**Competing interests** None declared.

**Patient consent for publication** Parental/guardian consent obtained.

**Ethics approval** Research Ethics Committee of Capital Institute of Pediatrics of China.

**Provenance and peer review** Not commissioned; externally peer reviewed.

**Data sharing statement** Extra data is available by emailing Jiemi@vip.163.com.

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
