## [Reviewer comments · BMJ Open]

ARTICLE DETAILS

TITLE (PROVISIONAL)	Bone mineral density reference standards for Chinese children aged 3-18: Cross-sectional results of the 2013-2015 China Child and Adolescent Cardiovascular Health (CCACH)
AUTHORS	Liu, Juntong; Wang, Liang; Sun, Jinghui; Liu, Gongshu; Yan, Weili; Xi, Bo; Xiong, Feng; Ding, Wenqing; Huang, Guimin; Heymsfield, Steven; Mi, Jie

VERSION 1 - REVIEW

REVIEWER	An Pan School of Public Health, Tongji Medical College, Huazhong University of Science and Technology, China
REVIEW RETURNED	15-Jan-2017

GENERAL COMMENTS	In this manuscript, Liu et al used data from a large nationwide cross-sectional study and provided sex-specific BMD reference values for Chinese children and adolescents (3-18 years). The findings have significant public health and clinical importance. Overall, the manuscript was well written and the analyses seemed appropriate. I have the following minor comments: 1. It will be better to have a study flow chart to show how many children were eligible, how many were invited, and how many agreed, in each center. To derive the reference curve, representativeness is a very important factor.2. The description of the quality control procedure was ok. To compare the results directly with data from US, did the author compare the procedures or protocol (including machine) between your study and the US study?3. The authors used several measures (TBLH BMD, BMD, and BMD by height) in the paper, to individuals who are not familiar with the terminology and clinical meanings of those measures, the results and findings are difficult to follow. For example, in the Results section of the Abstract, the authors described: "Furthermore, TBLH BMD was higher among U.S. children as compared to Chinese children. There was a large difference in BMD for height among children from the two countries."4. When comparing the results between your study and the US study, it is unclear why the authors specifically discussed P3, P50, and P97, why not P10, P50, P90? Any rationale?5. In the Results section, many numbers were shown and they were quite difficult to be remembered and understood. The results section
--

	should present the major findings without replicating too many numbers from the tables and figures. 6. Table 1 was not described in the manuscript. What is the meaning of CV% of those measures? How were they used in the paper? The first sentence in the Results section was about Table 2, not Table 1. 7. In Table 2, the TBLH BMD was shown for only 6 centers? 8. In the supplemental tables, the L was the same for the males, but not the females, any reasons? 9. Page 3, under “Strengths and limitations of this study”, it should be “Data we collected were”, not “Data we collected is”
--	--

REVIEWER	Dianjianyi Sun Tulane University, USA
REVIEW RETURNED	06-Feb-2017

GENERAL COMMENTS	This is a well-written, carefully performed study which constructed solid reference curves and extended previous studies though covering a wide-range age from 3 to 18 based on a large sample size in 7 areas in China. There are some remaining questions. Five major concerns:  1. A strong introduction is needed regarding pediatric osteoporosis, growth, or early life development, rather than too much on osteoporosis in the elder. 2. Can the authors expand on public health importance and clinical implications? 3. Is there any explanation for the turning point at 14 yrs. of age and 155 cm of height in girls? 4. Given the significant difference among six areas shown in Table 2, as well as the non-representativeness, how to control the geographical heterogeneity? Have you considered using a mixed model or meta-analysis to generate the point estimates specific for age, gender, and height? 5. Which US pediatric population you used for comparison, white, blacks, or Mexican Americans? As all the results (e.g. the Table S13 and Table S17) reported stratified by three ethnicities in the paper entitled “Dual Energy X-Ray Absorptiometry Body Composition Reference Values from NHANES”. 6. One paper, with the title of “Age trends of bone mineral density and percentile curves in healthy Chinese children and adolescents”, also performed the same analysis in southern Chinese pediatric population. Comparison with the above study is also needed, and better for your interpretation and explanation of the area heterogeneity. 7. Why the parameter “L” is all the same in boys in Table S1 and S2? Minor comments:  1. Different genetic background should also be a primary explanation for the difference in BMD among children in China and US. 2. What else the results provided by DXA, fat mass, lean mass, total body BMD, % fat mass, and lean Mass/Height², et al.? 3. Paragraph 2 in the discussion is more suitable moving to the introduction. 4. A mistake in“.....Ningxia accounts for the smallest proportion of
---

	12.2% (Table 1)", it's actually Table 2. 5. "DXA" in the "introduction" part should move to methods. 6. "%" inside Table 1 should be deleted. 7. Footnotes for abbreviations in Tables were required in Table S1 and S2. 8. Figure legends were missing.
--	---

VERSION 1 – AUTHOR RESPONSE

Reviewer: 1

Reviewer Name: An Pan

Institution and Country: School of Public Health, Tongji Medical College, Huazhong University of Science and Technology, China

Competing Interests: None declared.

In this manuscript, Liu et al used data from a large nationwide cross-sectional study and provided sex-specific BMD reference values for Chinese children and adolescents (3-18 years). The findings have significant public health and clinical importance. Overall, the manuscript was well written and the analyses seemed appropriate.

I have the following minor comments:

1. It will be better to have a study flow chart to show how many children were eligible, how many were invited, and how many agreed, in each center. To derive the reference curve, representativeness is a very important factor.

Response:

We added a flow chart. Please see Figure 1.

2. The description of the quality control procedure was ok. To compare the results directly with data from US, did the author compare the procedures or protocol (including machine) between your study and the US study?

Response:

Thanks for the suggestion of the reviewer. It is essential to select norms collected by using equipment from the same manufacturer as that used for the participants due to systematic differences in software (Crabtree et al., 2014). Our results were compared with U.S. studies that used DXA data of U.S. National Health and Nutrition Examination Survey (NHANES). Consistent with NHANES data, we collected DXA data using the same machine and followed procedures recommended by the International Society for Clinical Densitometry (ISCD). In addition, all staff were trained by ISCD (see details in the quality control section).

Reference

Crabtree NJ, Arabi A, Bachrach LK, et al; International Society for Clinical Densitometry. Dual-energy X-ray absorptiometry interpretation and reporting in children and adolescents: the revised 2013 ISCD Pediatric Official Positions. *J Clin Densitom.* 2014;17(2):225–242

3. The authors used several measures (TBLH BMD, BMD, and BMD by height) in the paper, to individuals who are not familiar with the terminology and clinical meanings of those measures, the results and findings are difficult to follow. For example, in the Results section of the Abstract, the

authors described: “Furthermore, TBLH BMD was higher among U.S. children as compared to Chinese children. There was a large difference in BMD for height among children from the two countries.”

Response:

We added more description/explanation for those who may not be familiar with the terminology.

Bone mass, as measured by DXA, is reported as BMC (g) or areal BMD (g/cm²). These values are compared with reference values from healthy youth of similar age, sex, and race/ethnicity to calculate a z score, the number of SDs from the expected mean (Bachrach et al., 2016). The preferred skeletal sites for DXA measurements in children are lumbar spine (L1–4) and total body, not including the head (Crabtree et al., 2014). The cranium should be excluded from the total body scan analysis, because the head constitutes a large portion of the total body bone mass but changes little with growth, activity, or disease; inclusion of the skull potentially masks gains or losses at other skeletal sites (Taylor et al., 1997). Height and age are important indicators for evaluating child’s growth. We have obtained results of TBLH BMD for age and TBLH BMD for height, and compared with those of NHANES.

References

Bachrach LK, Gordon CM, SECTION ON ENDOCRINOLOGY. Bone densitometry in children and adolescents. *Pediatrics*. 2016; 138(4). pii: e20162398.

Crabtree NJ, Arabi A, Bachrach LK, et al; International Society for Clinical Densitometry. Dual-energy X-ray absorptiometry interpretation and reporting in children and adolescents: the revised 2013 ISCD Pediatric Official Positions. *J Clin Densitom*. 2014;17(2):225–242

Taylor A, Konrad PT, Norman ME, Harcke HT. Total body bone mineral density in young children: influence of head bone mineral density. *J Bone Miner Res*. 1997;12(4):652–655

4. When comparing the results between your study and the US study, it is unclear why the authors specifically discussed P3, P50, and P97, why not P10, P50, P90? Any rationale?

Response:

These percentiles give an indication of the percentage of the concerning population that is below a certain value. P3, P50, and P97 approximately refer to -2SD, Mean, and +2SD; while P3 and P97 are commonly used as low and high percentile in BMD research and other children’s growth and development indicators. They were used in this study to facilitate comparison with other studies.

5. In the Results section, many numbers were shown and they were quite difficult to be remembered and understood. The results section should present the major findings without replicating too many numbers from the tables and figures.

Response:

We have revised accordingly to make it more concise, e.g., presenting the major findings. See pages 10 and 11.

6. Table 1 was not described in the manuscript. What is the meaning of CV% of those measures? How were they used in the paper? The first sentence in the Results section was about Table 2, not

Table 1.

Response:

We described Table 1 in the section of "DXA acquisition and quality control". Table 1 shows results of precision quality control required by ISCD conducted by all our operator staff. CV% refers to the coefficient of variation for repeated measures by operator staff. As stated in main text, see below.

"All our operator staff passed the precision quality control required by International Society for Clinical Densitometry (ISCD), which controlled the coefficient of variation (CV %) to less than 1.9% for lumbar spine scans, 1.8% for hip joint scans, and 2.5% for femoral neck scans (Baim et al., 2005) (see Table 1)".

Reference

Baim S, Wilson CR, Lewiecki EM, Luckey MM, Downs RW, Lentle BC. Precision assessment and radiation safety for dual-energy X-ray absorptiometry: position paper of the International Society for Clinical Densitometry. *J Clin Densitom.* 2005. 8(4): 371-8.

7. In Table 2, the TBLH BMD was shown for only 6 center s?

Response:

Children were recruited from five geographical areas of North, East, Northeast, Northwest, and Southwest of China, including seven centers in Jilin Province (Northeast), Beijing and Tianjin (North), Ningxia Province (Northwest), Shanghai and Shandong Province (East), and Chongqing (Northwest). Table 2 shows TBLH BMD for these five geographical areas.

8. In the supplemental tables, the L was the same for the males, but not the females, any reasons?

Response:

The LMS technique (Cole and Green, 1992; Pan and Cole, 2004) estimates three parameters: median (M), coefficient of variation (S), and power in the Box-Cox transformation (L). This curve fitting procedure is able to normalize the underlying reference data by dividing the independent measure (e.g. age) into groups and then applying a power transformation, eliminating skewness in the variable under analysis. A smooth curve is fitted to the normalizing power transformation for each age group, generating an optimum "L" (power) curve that normalizes the dependent measure over the entire age range. "L" values may vary (i.e., different or the same) by the distribution of original data. Many other studies (Kang et al., 2016; Kelly et al, 2009) have obtained similar results.

References

Kelly TL, Wilson KE, Heymsfield SB. Dual energy X-Ray absorptiometry body composition reference values from NHANES. *PLoS One.* 2009. 4(9): e7038.

Kang MJ, Hong HS, Chung SJ, Lee YA, Shin CH, Yang SW. Body composition and bone density reference data for Korean children, adolescents, and young adults according to age and sex: results of the 2009-2010 Korean National Health and Nutrition Examination Survey (KNHANES). *J Bone Miner Metab.* 2016. 34(4): 429-39.

9. Page 3, under "Strengths and limitations of this study", it should be "Data we collected were",

not "Data we collected is"

Response:

It has been revised.

Reviewer: 2

Reviewer Name: Dianjianyi Sun

Institution and Country: Tulane University, USA

Competing Interests: None declared

This is a well-written, carefully performed study which constructed solid reference curves and extended previous studies though covering a wide-range age from 3 to 18 based on a large sample size in 7 areas in China. There are some remaining questions.

Five major concerns:

1. A strong introduction is needed regarding pediatric osteoporosis, growth, or early life development, rather than too much on osteoporosis in the elder.

Response:

We have revised as suggested by the reviewer.

Pediatric osteoporosis is defined as having a history of pathologic fractures and low bone mineral content or density (Uziel et al., 2009). To diagnose pediatric osteoporosis, DXA is the most widely used bone densitometry technique in children (Uziel et al., 2009). Although 60% of peak bone mass is genetically determined, the increase of bone mass can be influenced by other factors, including intake of dietary products (e.g., calcium and vitamin D) and weight-bearing physical activity (e.g., jumping) (Rizzoli et al., 2010). While diagnosing children can be problematic due to the lack of a widely used reference, the Bone Mineral Density in Childhood Study (BMDCS) has developed an online BMD z-score calculator for health care providers (<https://bmdcs.nichd.nih.gov/zscore.htm>). Widespread reference for multiple diagnostic measurements is important to ensure correct diagnosis among children so that treatment and management for osteoporosis can begin. Uniformity in diagnosis is also imperative for public health so that epidemiologic studies can be done across geographic regions, ethnicities, and social economic status to properly inform prevention practices. Moon et al. (2016) studied the accuracy of parental recall of children's fractures. They determined that, of 660 parents, 207 children had previous fractures. Of these, 21% were reported incorrectly, indicating a need to have a reporting system for fractures so as to better identify osteoporosis in children (Moon et al., 2016). A study published in 2014, created reference data for GE Lunar systems from the U.S. National Health and Nutrition Examination Survey (NHANES) (Fan et al.). To do this they took data from 8056 participants under the age of 20 and created reference curves by age, sex, and ethnicity (Fan et al., 2014). Kinning et al. discovered that the rearrangement of chromosomes 4:20 is associated with childhood osteoporosis, highlighting the need to look further into genetic predispositions for diseases (2016).

Childhood and adolescence are critical periods for bone growth, while half of peak bone mass is developed in childhood. If BMD could be evaluated in childhood and adolescence, and those with low BMD could be intervened, peak bone mass can be improved obviously in adulthood, osteoporosis occurrence can be delayed, and severity of osteoporosis can be reduced. A number of studies have shown that optimal BMC and BMD in childhood and adolescence will reduce risk of osteoporotic

fracture several decades later (NIH, 2001).

References

Fan B, Shepherd JA, Levine MA, et al. National Health and Nutrition Examination Survey whole-body dual-energy X-ray absorptiometry reference data for GE Lunar systems. *J Clin Densitom.* 2014;17(3):344-377.

Kinning E, McMillan M, Shepherd S, et al. An unbalanced rearrangement of chromosomes 4:20 is associated with childhood osteoporosis and reduced caspase-3 levels. *J Pediatr Genet.* 2016; 5(3): 167-173.

Moon RJ, Lim A, Farmer M, et al. Validity of parental recall of children's fracture: implications for investigation of childhood osteoporosis. *Osteoporos Int.* 2016; 27(2): 809–813.

NIH Consensus Development Panel on Osteoporosis Prevention, Diagnosis, and Therapy. Osteoporosis prevention, diagnosis, and therapy. *JAMA.* 2001; 285:785–795.

Rizzoli R, Bianchi ML, Garabédian M, McKay HA, Moreno LA. Maximizing bone mineral mass gain during growth for the prevention of fractures in the adolescents and the elderly. *Bone.* 2010; 42(2): 294-305.

Uziel Y, Zifman E, & Hashkes PJ. (2009). Osteoporosis in children: pediatric and pediatric rheumatology perspective: a review. *Pediatr Rheumatol Online J.* 2009; 7:16.

2. Can the authors expand on public health importance and clinical implications?

We added public health importance and clinical implications as suggested by the reviewer. Please see below.

There are some important public health implications. First, the BMD percentile can be used to precisely evaluate child and adolescent's growth, and help high-risk population for better intervention. Second, since childhood obesity has been epidemic in the nation, assessing BMD can facilitate association studies between body composition (e.g., fat, muscle, etc.) and bone health, to better determine health status for obese individuals. Furthermore, it can be used for evaluating effectiveness of sports intervention among children. Clinical implications are also notable. First, child's kidney disease, rheumatism and immune disease, bone and joint disease can directly and indirectly influence bone growth; while BMD test can evaluate such diseases' adverse outcome on individual's health status and treatment effect. In addition, antiepileptic drugs and other hormonal drugs can affect osteogenesis, and BMD reference can be used as a standard for evaluation of treatment effect and side effects for children.

2. Is there any explanation for the turning point at 14 yrs. of age and 155 cm of height in girls?

Response:

In our research, the average height for girls was 160.0 ± 6.2 cm at 14 years, during which changes of BMD may be associated with pubertal development. The present study has not collected data in puberty, which can be investigated in future studies.

4. Given the significant difference among six areas shown in Table 2, as well as the non-representativeness, how to control the geographical heterogeneity? Have you considered using a

mixed model or meta-analysis to generate the point estimates specific for age, gender, and height?

Response:

Mixed model can do stratified analysis for outcome variables after controlling for covariates, and obtain adjusted percentiles for outcome variables. However, the assumption of mixed model is to have the normal distribution of outcome variables; if not, normal distribution needs to be transformed. LMS has been used in the present study. The LMS technique (Cole and Green, 1992; Pan and Cole, 2004) estimates three parameters: median (M), coefficient of variation (S), and power in the Box-Cox transformation (L). This curve fitting procedure is able to normalize the underlying reference data by dividing the independent measure (e.g. age) into groups and then applying a power transformation, eliminating skewness in the variable under analysis. A smooth curve is fitted to the normalizing power transformation for each age group, generating an optimum “L” (power) curve that normalizes the dependent measure over the entire age range, which cannot be completed by mixed model.

5. Which US pediatric population you used for comparison, white, blacks, or Mexican Americans? As all the results (e.g. the Table S13 and Table S17) reported stratified by three ethnicities in the paper entitled “Dual Energy X-Ray Absorptiometry Body Composition Reference Values from NHANES”.

Response:

We compared our results to U.S. whites. We have revised to make it more clearly.

6. One paper, with the title of “Age trends of bone mineral density and percentile curves in healthy Chinese children and adolescents”, also performed the same analysis in southern Chinese pediatric population. Comparison with the above study is also needed, and better for your interpretation and explanation of the area heterogeneity.

Response:

Compared to our study, another study (Guo et al., 2013) conducted among southern Chinese pediatric population is different in selected geographic characteristics, sample size, and age range. For example, although 900 children (5-19 years) came from Guangzhou (Southern China) and about 600 (14-19 years) came from Zhejiang (East China), the study did not consider variation in geographic characteristics, and age range was smaller than ours (3-18 years). Equipment used in that study was GE Lunar, the results of which cannot be compared with ours using Hologic DXA.

Reference

Guo B, Xu Y, Gong J, Tang Y, Xu H. Age trends of bone mineral density and percentile curves in healthy Chinese children and adolescents. *J Bone Miner Metab.* 2013 May;31(3):304-314.

7. Why the parameter “L” is all the same in boys in Table S1 and S2?

Response:

The LMS technique (Cole and Green, 1992; Pan and Cole, 2004) estimates three parameters: median (M), coefficient of variation (S), and power in the Box-Cox transformation (L). This curve fitting procedure is able to normalize the underlying reference data by dividing the independent measure (e.g. age) into groups and then applying a power transformation, eliminating skewness in the variable under analysis. A smooth curve is fitted to the normalizing power transformation for each age group, generating an optimum “L” (power) curve that normalizes the dependent measure over the entire age

range. “L” values may vary (i.e., different or the same) by the distribution of original data. Many other studies (Kang et al., 2016; Kelly et al., 2009) have obtained similar results as ours using the LMS method.

References

Cole TJ, Green PJ. Smoothing reference centile curves: the LMS method and penalized likelihood. *Stat Med.* 1992;11(10):1305-1309.

Kang MJ, Hong HS, Chung SJ, Lee YA, Shin CH, Yang SW. Body composition and bone density reference data for Korean children, adolescents, and young adults according to age and sex: results of the 2009-2010 Korean National Health and Nutrition Examination Survey (KNHANES). *J Bone Miner Metab.* 2016. 34(4): 429-39.

Kelly TL, Wilson KE, Heymsfield SB. Dual energy X-Ray absorptiometry body composition reference values from NHANES. *PLoS One.* 2009. 4(9): e7038.

Pan H, Cole TJ. A comparison of goodness of fit tests for age-related reference ranges. *Stat Med.* 2004; 23(11):1749-1765.

Minor comments:

1. Different genetic background should also be a primary explanation for the difference in BMD among children in China and US.

Response:

We added this possible explanation.

2. What else the results provided by DXA, fat mass, lean mass, total body BMD, % fat mass, and lean Mass/Height², et al.?

Response:

We added additional supplemental tables, and provided comprehensive results (e.g., FMP, FMI, and FFMI) by DXA.

3. Paragraph 2 in the discussion is more suitable moving to the introduction.

Response:

We have revised as suggested by the reviewer.

4. A mistake in “.....Ningxia accounts for the smallest proportion of 12.2% (Table 1)”, it’s actually Table 2.

Response:

Done

5. “DXA” in the “introduction” part should move to methods.

Response:

We have revised as suggested by the reviewer.

6. “%” inside Table 1 should be deleted.

Response:

Done.

7. Footnotes for abbreviations in Tables were required in Table S1 and S2.

Response:

Done

8. Figure legends were missing.

Response:

We added figure legends after references.

VERSION 2 – REVIEW

REVIEWER	An Pan School of Public Health, Tongji Medical College, Huazhong University of Science and Technology, China
REVIEW RETURNED	09-Mar-2017

GENERAL COMMENTS	The authors have addressed most of my concerns. However, the revised Introduction is too long (almost three pages), and should be substantially condensed to focus on the studies on this topic. For example, the Moon study was not relevant, so as the study with reference No. 10 and 18 thereafter. In addition, reference 10 study was cited as a study using DXA in the second paragraph, then it was described as a study for "GE Lunar systems" in the fourth paragraph. Anyway, the introduction should focus on study of establishing reference standards for childhood and adolescence in different populations, why it is important to have the reference data in China etc; other irrelevant sentences should be deleted or moved to the Discussion section. For figure 1, from 15548 to 11457, more than 4000 participants were excluded, while you only gave reasons for 109 and 130 participants as shown in the figure. For table 1, it should be clearly described that the CV% was for the quality control procedure, not the CV% for the final collected data, as you mentioned in your Methods section.
---

REVIEWER	Dianjianyi Sun Tulane University, USA
REVIEW RETURNED	12-Mar-2017

GENERAL COMMENTS	All the comments were well addressed by the authors.
--

VERSION 2 – AUTHOR RESPONSE

Reviewer: 1

Reviewer Name: An Pan

Institution and Country: School of Public Health, Tongji Medical College, Huazhong University of Science and Technology, China Competing Interests: None declared

The authors have addressed most of my concerns. However, the revised Introduction is too long (almost three pages), and should be substantially condensed to focus on the studies on this topic. For example, the Moon study was not relevant, so as the study with reference No. 10 and 18 thereafter. In addition, reference 10 study was cited as a study using DXA in the second paragraph, then it was described as a study for "GE Lunar systems" in the fourth paragraph. Anyway, the introduction should focus on study of establishing reference standards for childhood and adolescence in different populations, why it is important to have the reference data in China etc; other irrelevant sentences should be deleted or moved to the Discussion section.

Response:

We have revised as suggested by the reviewer. See page 5.

For figure 1, from 15548 to 11457, more than 4000 participants were excluded, while you only gave reasons for 109 and 130 participants as shown in the figure.

Response:

We added more details to make it more clearly.

Among the 15,548 eligible participants, 11,457 agreed to participate and signed written informed consent documents. Data were collected on 11,457 children; of these 400 Hui children were excluded, 109 were excluded due to kidney or thyroid disease, and 130 were excluded because of implausible values of weight (light or heavy, BMI ≥ 3 SDs). Consequently, 10,818 children (5,509 males and 5,309 females) were used in the final analysis. The figure 1 has been revised.

For table 1, it should be clearly described that the CV% was for the quality control procedure, not the CV% for the final collected data, as you mentioned in your Methods section.

Response:

We revised as suggested.

There are two places that mentioned CV% in Methods section, which are different. One is for the quality control procedure, which measures coefficient of variation for DXA equipment used for collecting data; the other refers to coefficient of variation for how operator staff perform precision quality control required by the International Society for Clinical Densitometry.

Reviewer: 2

Reviewer Name: Dianjianyi Sun

Institution and Country: Tulane University, USA Competing Interests: None declared

All the comments were well addressed by the authors.

Response:

Thank you.

VERSION 3 – REVIEW

REVIEWER	An Pan Huazhong University of Science and Technology
REVIEW RETURNED	20-Mar-2017

GENERAL COMMENTS	The reviewer completed the checklist but made no further comments.
--

REVIEWER	Dianjianyi Sun Epidemiology, Tulane University
REVIEW RETURNED	20-Mar-2017

GENERAL COMMENTS	The authors have done a very nice job of responding to the reviewers clarifying the presentation.
---